# Maternal vitamin B₁₂, folate during pregnancy and neurocognitive outcomes in young adults of the Pune Maternal Nutrition Study (PMNS) prospective birth cohort: study protocol

Rishikesh V Behere [ID],[1] Gopikrishna Deshpande,[2,3,4] Souvik Kumar Bandyopadhyay,[5] Chittaranjan Yajnik [ID][1]

For numbered affiliations see end of article.

**Correspondence to**
Dr Rishikesh V Behere;
rvbehere@gmail.com

## ABSTRACT

**Introduction** The Developmental Origins of Health and Disease (DOHaD) hypothesis proposes that intrauterine and early life exposures significantly influence fetal development and risk for disease in later life. Evidence from prospective birth cohorts suggests a role for maternal B₁₂ and folate in influencing neurocognitive outcomes in the offspring. In the Indian setting, B₁₂ deficiency is common during the pregnancy while rates of folate deficiency are lower. The long-term influences of maternal nutrition during the pregnancy on adult neurocognitive outcomes have not been examined. The Pune Maternal Nutrition Study (PMNS) is a preconceptional birth cohort into its 24th year and is considered a unique resource to study the DOHaD hypothesis. We found an association between maternal B₁₂ status in pregnancy and child's neurocognitive status at 9 years of age. We now plan to assess neurocognitive function and MRI measurements of brain structural–functional connectivity at young adult age to study its association with maternal nutritional exposures during the pregnancy.

**Methods and analysis** As part of ongoing prospective follow-up in young adults of the PMNS at the Diabetes Unit, KEM Hospital Research Center, Pune India, the following measurements will be done: neurocognitive performance (Standardised Tests of Intelligence, Verbal and Visual Memory, Attention and Executive Functions), temperament (Adult Temperament Questionnaire), psychopathology (Brief Symptom Inventory and Clinical Interview on Mini Neuropsychiatric Interview 7.0). Brain MRI for structural T1, resting-state functional connectivity and diffusion tensor imaging will be performed on a subset of the cohort (selected based on exposure to a lower or higher maternal B₁₂ status at 18 weeks of pregnancy).

**Ethics and dissemination** The study is approved by Institutional ethics committee of KEM Hospital Research Center, Pune. The results will be shared at national and international scientific conferences and published in peer-reviewed scientific journals.

**Trial registration number** NCT03096028

## Strengths and limitations of this study

► Investigation of structural and functional neurodevelopment of young adults in the Pune Maternal Nutrition Study in relation to intrauterine exposure to vitamin B₁₂ and folate.
► A unique dataset of intergenerational life course exposures (physical growth, nutrition, childhood adversity) to investigate pathways of neurocognitive development.
► Comprehensive behavioural measures and neuroimaging techniques (structural and functional).
► Cross-sectional MRI measurements instead of trajectories of brain growth.
► First life course study in Low and Middle Income Countries (LMIC) to evaluate associations of early life exposures with brain outcomes in adult age.

## INTRODUCTION

The Developmental Origins of Health and Disease (DOHaD) hypothesis[1 2] proposes that early life exposures along with postnatal environmental factors (childhood experiences, education, nutrition, infections, lifestyle, socioeconomic conditions) determine risk for disease in later life. Recent investigations suggest an early life origin of neurodevelopmental disorders (autism spectrum disorders, depression, schizophrenia).[3] Early intrauterine insults can impact development of brain areas that show high neuronal growth such as hippocampus and cortical areas.[4] Later life factors influence synaptic pruning and myelination.[5] Altered brain structural and functional connectivity within and between hippocampal, frontal and subcortical networks is known to be associated with impaired neurocognitive functioning and neurodevelopmental disorders.[6 7]

The growing fetus is wholly dependent on the mother for its nutrition. Deficiency of various micronutrients in the mother during critical phases of fetal development can affect organ structure and function, hence programming the risk for chronic disease in later life (fetal programming).[8] Recently, there is considerable interest in the role of maternal vitamin $B_{12}$ and folate in fetal programming.

Vitamin $B_{12}$ is important for nerve myelination and its deficiency is associated with subacute combined degeneration of spinal cord, depression and dementia.[9] The role of folate deficiency in risk for neural tube defects (NTDs) is well known. Both these vitamins influence the one carbon metabolism cycle.[10]

Observations from large prospective cohort studies such as the Norwegian mother and child birth cohort[11] and Generation R support association of poor maternal folate status with poorer neurocognitive functioning, greater childhood behavioural problems,[12–14] smaller brain volumes[15] and risk for autism.[16 17] Prospective cohort studies from India support the associations of lower maternal vitamin $B_{12}$ or folate with higher risk for NTD and poorer cognitive functions in offspring in early (30–48 months) and late (9–10 years) childhood. Systematic reviews support a high level of evidence for association of low maternal vitamin $B_{12}$ with high risk for NTD and moderate level evidence for poor cognitive outcomes in the offspring.[18 19] It must be noted that these observational studies do not support causality. Causality can be examined by randomised controlled trials, which need time and involve ethical considerations. Techniques such as Mendelian randomisation allow testing for causality of observed associations.[20] The Avon Longitudinal Study on Parents And Children reported association between low maternal dietary intake of vitamin $B_{12}$ and lower IQ scores at 8 years and genetic determinants of vitamin $B_{12}$ (FUT2gene) using this technique.[21]

In India, vitamin $B_{12}$ deficiency is widely prevalent in women in the reproductive age group while rates of folate deficiency are low.[22–24] The vitamin $B_{12}$ deficiency is attributable to practice of vegetarianism and low socioeconomic status. Public health policy mandates only iron-folic acid supplementation in adolescents and pregnant women. It is also a common clinical practice to supplement high dose folic acid (5 mg) to pregnant women to prevent NTD. This scenario inadvertently promotes an imbalance between vitamin $B_{12}$ and folate status during the pregnancy,[19 25] which could enhance risk of adverse health outcomes in the offspring including aberrant neurodevelopment. In the presence of vitamin $B_{12}$ deficiency, folate is trapped in its unusable methyl form and methyl groups are unavailable for further biochemical reactions. This interferes with important cell processes such as purine pyrimidine synthesis, DNA repair and epigenetic regulation (DNA methylation). Unavailability of methyl groups prevents conversion of homocysteine to S-adenosyl methionine leading to hyper-homocysteinaemia which can adversely affect vascular endothelial and placental function. Indiscriminate supplementation in the presence of vitamin $B_{12}$ deficiency leads to accumulation of unmetabolised folic acid, which is known to have neurotoxic effects.[25 26] Animal experiments performed in India have examined the role of a maternal vitamin $B_{12}$-folate micronutrient imbalance and show that offspring dams of mothers fed on a low vitamin $B_{12}$-highfolate diet have poorer spatial memory functions, lower brain hippocampal weight, reduced neurotrophic factor BDNF and reduced TRK gene expression.[27 28] However, no study in India has examined long-term effects of maternal vitamin and folate during the pregnancy on offspring neurodevelopmental outcomes in young adulthood.

The Pune Maternal Nutrition Study (PMNS) is a preconceptional birth cohort with serial data on maternal nutritional status, birth outcomes and offspring nutrition, physical growth and development. We have observed high prevalence of maternal vitamin $B_{12}$ deficiency during the pregnancy in our cohort. We have described markers of low vitamin $B_{12}$ status in the mother (high homocysteine) to be causally associated with low birth weight in the offspring.[29] A low vitamin $B_{12}$-high folate dietary pattern was associated with adverse metabolic outcomes in the offspring at 6 years.[24] We have earlier demonstrated in this cohort that exposure to a low maternal vitamin $B_{12}$ was associated with poor cognitive functioning in the offspring at 9 years of age.[30] The offspring of this cohort is now in their early 20s and this provides a unique opportunity to examine the long-term effects of maternal nutrition status during pregnancy on neurocognitive outcomes in young adulthood.

## Aims and objectives
The key aim of the study is to examine whether maternal micronutrient status during pregnancy ($B_{12}$, folate) is associated with neurocognitive performance, structural and functional connectivity and risk for common mental disorders in young adult offspring of the PMNS birth cohort.

Primary objectives
1. To examine associations between maternal vitamin $B_{12}$, folate and their interaction at 18 and 28 weeks of pregnancy with neurocognitive outcomes (neuropsychological test performance) and risk for common mental disorders (temperament, presence of depression or anxiety disorders) in young adulthood; controlling for the effects of other nutrients (calorie, protein and fat intake, and levels of vitamin C, D, ferritin) and later life developmental influences (education, adverse childhood experiences (ACE), offspring nutrition, physical growth and development, metabolic status).
2. To examine the association of maternal nutritional status during the pregnancy with brain structure (grey matter volume and cortical thickness), and structural/functional connectivity (using diffusion tensor imaging (DTI) and resting state functional MRI (rsfMRI), respectively).
3. To test the causality of association between maternal vitamin $B_{12}$, folate and neurocognitive outcomes using

maternal and offspring genetic determinants of vitamin $B_{12}$ (FUT2, FUT6, TCN2).

## Hypotheses

We hypothesise that a lower maternal vitamin $B_{12}$, higher folate, higher homocysteine nutrient pattern during pregnancy adversely impacts development of brain regions. This will be reflected in brain imaging as reduced brain volume as measured through anatomical images, weaker connections in structural/functional networks (hippocampal and frontal-subcortical connections) involved in cognitive processing (memory, executive functions) as measured through DTI and rsfMRI, respectively. These structural and functional changes will in turn result in poorer neurodevelopmental outcomes (lower neuropsychological test performance, adverse temperamental traits and higher risk for common mental disorders) in the young adult offspring.'

## METHODS AND ANALYSIS

### Study setting

#### Pune Maternal Nutrition Study

The PMNS is a preconceptional cohort, established in six villages of Pune district, Maharashtra state, India in 1993 in collaboration with Dr DJP Barker and Dr Caroline Fall and is considered a unique resource in the world to examine the DOHaD paradigm. Initially, 2466 women (F0 generation) in the reproductive age group participated in the study. A total of 814 women who became pregnant in the period June 1994 to April 1996 were enrolled for the study. Of these 797 pregnancies (with singleton pregnancy and period of gestation less than 21 weeks) were included. A total of 762 live births occurred (F1 generation). This cohort has data on preconceptional parental size, maternal nutrition and size at birth of the offspring. Maternal micronutrients vitamin $B_{12}$, red cell folate and ferritin have been measured from blood samples collected at 18 and 28 weeks of gestation. Vitamin $B_{12}$ levels were measured using the microbiological assay technique and red cell folate and ferritin by radioimmunoassay.[24 31] Children are being regularly followed up for growth and development (body size and DXA body composition), nutrition and cardiometabolic risk factors (6, 12 and 18 years of age). Biological samples have been collected and stored in a biorepository and are available for assessments. The other strengths of this cohort are good community participation and follow-up rates of over 90% over the years. At 18 years follow-up, 690 of the original 762 live births in the cohort were followed up.

The participants (F1 generation) of the cohort are also part of a randomised controlled trial (Pune Rural Intervention in Young Adolescents, PRIYA)[32] to examine intergenerational effects of micronutrient supplementation on various health outcomes. In 2012 at the age of 17 years 557 eligible members were randomised as part of the nutrition intervention trial into 1 of three groups (1) 2 micrograms/day of vitamin $B_{12}$, (2) 2 µg/day of vitamin $B_{12}$ with UNIMAPP multivitamins and 20 g milk powder and (3) placebo. The trial was completed in the boys in December 2017 and in the girls in September 2019. The effects of later life intervention will be adjusted during statistical analysis (figure 1—depicts time line of measurements in the PMNS cohort).

### Subject selection

All subjects of the PMNS cohort (n=690) will be invited for participation in the study (figure 2) subject to following inclusion criteria: (1) Maternal vitamin $B_{12}$, folate levels measured during pregnancy (2) Serial measurements done at 6, 12 and 18 years of age. Neurocognitive assessments, temperament and mental health assessments will be performed in all consenting subjects. Brain MRI will be performed on a subset of the cohort selected based on exposure to maternal vitamin $B_{12}$ levels at 18 weeks of gestation. Median vitamin $B_{12}$ level at 18-week gestation was 153 pM (103 175 pM being the first and fourth quartiles). Description of maternal characteristics at 18-week and 28-week gestation and offspring birth characteristics are provided in online supplemental material 1. Of the 690 participants still in follow-up, 152 subjects (78 males, 74 females) whose mothers had plasma vitamin $B_{12}$ concentration <103 pM and 154 subjects (88 males, 66 females) whose mothers had plasma vitamin $B_{12}$ concentration >175 pM will be eligible for participation in the brain MRI scans. One hundred participants each from these two groups (total of 200 subjects) will be randomly selected.

### Assessments

The exposures and outcome measurements are summarised in table 1.

#### Behavioural assessments

These include (1) Standardised neurocognitive test batteries—Wechsler's Adult Intelligence Scale-IV, Wechsler Memory Scale, Auditory Verbal Learning Test, Colour Trail Test, (2) Adult Temperament Questionnaire, (3) Mental health assessments-Brief Symptom Inventory, Mini Neuropsychiatric Interview (plus 7.0). Further details of behavioural assessments are provided in online supplemental material 2 and (4) Early life stress-WHO ACE International Questionnaire (ACE-IQ).

### MRI

Anatomical data using structural 3 dimensional (3D) T1-weighted MPRAGE (magnetisation prepared rapid gradient echo) sequence, rsfMRI with 2D gradient-echo EPI (echo planar imaging) sequence and DTI using readout segmentation of long variable echo-trains (RESOLVE) sequence will be acquired on a 3 Tesla Siemens Skyra scanner.

### Image acquisition

For structural imaging, 3D T1-weighted MPRAGE images will be acquired (TR=2300 ms, TE=2.26 ms, TI=900 ms, Flip angle=8 °, slice thickness=1 mm, in-plane

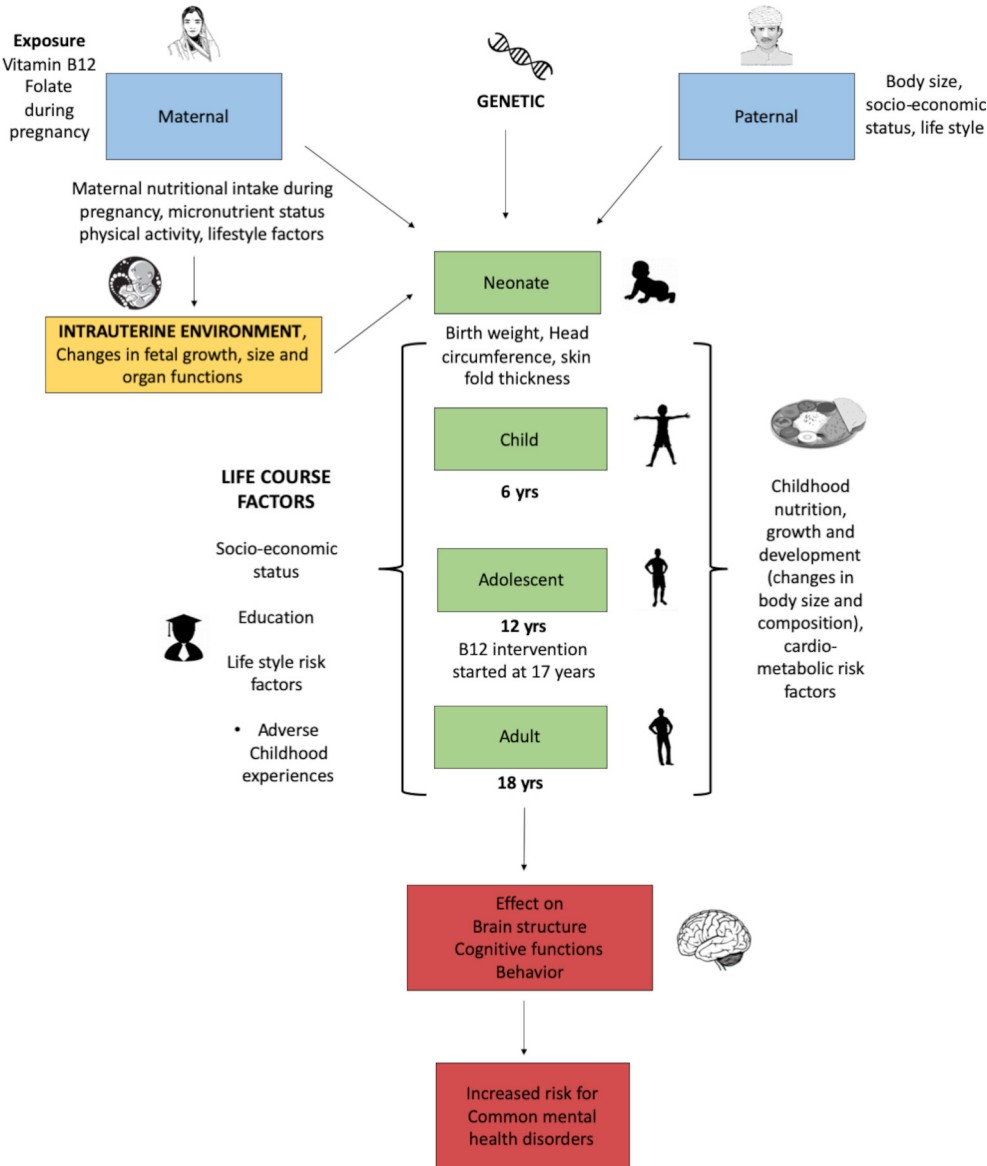

**Figure 1** Understanding neurocognitive development using a life course approach in the PMNS. PMNS, Pune Maternal Nutrition Study.

resolution=1×1 mm², GRAPPA parallel imaging with acceleration factor of, 192 slices, FOV readout=256 mm, FOV phase encoding=100%, and acquisition time of 5 min and 21 s).

For functional imaging, T2*-weighted echo-planar imaging sequence will be used to acquire BOLD fMRI data while participants are resting with eyes open and fixated, and not thinking about anything specific. A rsfMRI scan helps to characterise brain-wide networks from a single scan as opposed to task-based fMRI which characterises evoked responses in specific brain systems. Thirty-six axial slices will be acquired covering the whole brain with a slice thickness of 3 mm, in-plane resolution of 3×3 mm², TR=2000 ms; TE=30 ms; flip angle=90⁰, FOV readout=256 mm, FOV phase encoding=100%, GRAPPA parallel imaging with acceleration factor of, and 300 measurements with an acquisition time of 10 min and

8 s. Prospective acquisition correction (PACE) motion correction will be employed in this sequence.[33 34]

Diffusion-weighted images will be acquired using the RESOLVE sequence given its robustness against susceptibility artefacts. Scan parameters for RESOLVE will be: Thirty-six axial slices covering the whole brain with a slice thickness of 3 mm, in-plane resolution of 1.4×1.4 mm², TR=5460 ms; TE=73/109 ms; flip angle=180°, FOV readout=220 mm, FOV phase encoding=100%, GRAPPA parallel imaging with acceleration factor of 2, 30 diffusion directions, two diffusion weightings (b values) of 0 and 1000 s/mm² and an acquisition time of 14 min and 19 s.

All MRI images will be visually inspected for quality assessment and reviewed by a radiologist to rule out any structural abnormalities.

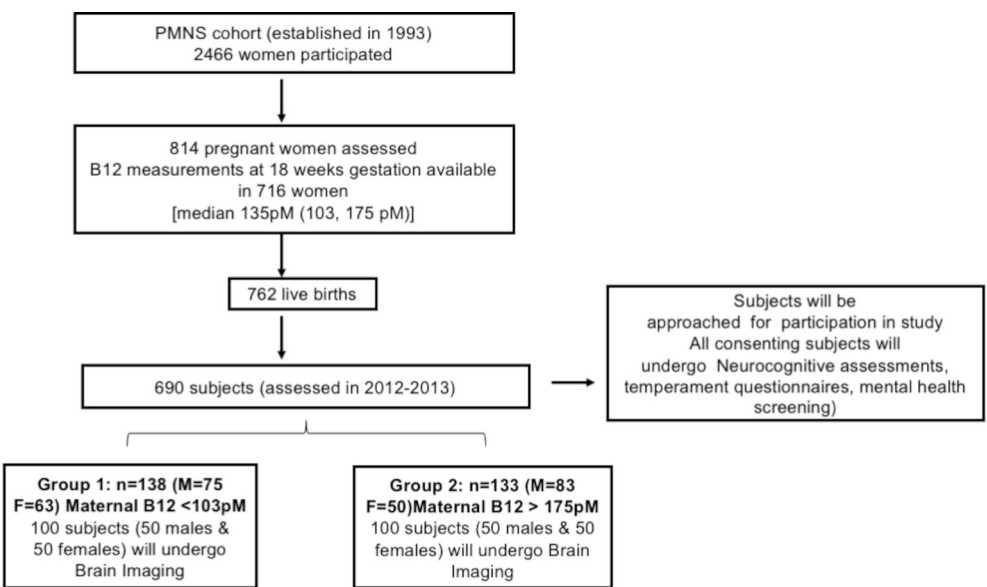

**Figure 2** Selection of subjects from the PMNS cohort for the study. PMNS, Pune Maternal Nutrition Study.

### Data management

Data will be stored at the Diabetes Unit, KEM Hospital Research Center, Pune, India. Data collected will be first entered in a paper clinical record form and double entered into an open clinical platform on a computer. Data will be downloaded and checked for extreme or missing values. Data on maternal micronutrient status during the pregnancy (nutrition intake, circulating

| | Table 1 Exposures and outcomes defined in the study | | |
|---|---|---|---|
| | **Exposures** | **Tests** | **Variables measured** |
| 1. | Maternal Micronutrient status and homocysteine at 18 and 28 weeks pregnancy | Biochemical assays | Vitamin $B_{12}$, red cell folate, homocysteine |
| | **Outcomes** | **Tests** | **Variables measured** |
| 1. | Neurocognitive performance | WAIS-IV WMS AVLT Colour Trails test | Verbal comprehension, perceptual reasoning, working memory, processing speed, Full scale IQ. Immediate memory, delayed memory and working memory. Errors in immediate and delayed recall time taken and errors for colour trails test A and B. |
| 2. | Temperament | Adult Temperament Questionnaire | Negative affect, extraversion, orienting sensitivity, effortful control |
| 3. | Common mental disorders | Brief Symptom Inventory Mini Neuropsychiatric Interview 7.0 for DSM 5 diagnosis | Global severity index DSM 5 diagnosis of past or current major depressive disorder, suicidality, panic disorder, agoraphobia, social anxiety disorder, generalised anxiety disorder |
| 5. | Brain Imaging | Structural T1 MPRAGE Resting state fMRI Diffusion tensor imaging | Cortical volume and thickness Resting state functional connectivity tractography, FA and MD values in relevant white matter tracts |
| | Life course variables | | |
| | Birth | | Birth weight, length, head circumference, sum of skin folds |
| | Later life growth and development | | Height, BMI, head circumference sum of skin folds at 6, 12, 18 years |
| | Later life nutrition | | Haemoglobin, vitamin $B_{12}$, folate, homocysteine, ferritin, glucose, lipids at 6,12,18 years |
| | Socioenvironmental | | Mothers education, fathers education, standard of living index, adverse childhood experiences. |

AVLT, auditory verbal learning test; BMI, body mass index; DSM, Diagnositic and Statistical Manual of Mental Disorders; FA, fractional anisotropy; fMRI, functional MRI; MD, mean diffusivity; WAIS-IV, Wechsler's Adult Intelligence Scale IV; WMS, Wechsler Memory Scale.

micronutrients) and intermediate variables (birth size, childhood nutrition, growth and development, metabolic factors (insulin resistance, glycaemic status, adiposity), sociodemographic variables and socioeconomic status have been assessed previously on the cohort. These will be extracted and used for analysis.

## Statistical methods
### Power estimation

There are 690 members of the cohort available for assessments. Assuming a 80% participation rate (from experience of previous studies on the cohort and accounting for migration) we estimate that 550 subjects would be recruited for the study. Using a test at 5% significance level, the study will have 80% power to detect an association of 0.11 SD of continuous outcome (eg, neurocognitive test score) per SD of continuous exposure (eg, maternal vitamin $B_{12}$ level). Study of 200 subjects (100 in each group) for brain imaging will have 80% power for an effect size 0.3 and significance at $p<0.05$ for detecting difference between the two maternal vitamin $B_{12}$ exposure groups.

### Current status and impact of COVID-19 pandemic

The study protocol was approved for funding in 2017. Subject recruitment was initiated in May 2018 and had to be halted in March 2020 due to the COVID-19 pandemic. As of March 2020, neurocognitive assessments have been performed on 386 participants of the cohort out of intended 550 participants and brain MRI scans obtained on 195 participants our of intended 200. We will try to recruit more participants once the situation is conducive. We acknowledge that due to the COVID-19 pandemic there will be some reduction in the power of the study. The results will be reported as per emerging guidelines on reporting clinical studies affected by the COVID-19 pandemic.[35]

## Plan of data analysis
### Objective 1

This examines the association between intrauterine nutritional exposures and neurocognitive outcomes in young adult age:

Our primary exposure of interest is maternal vitamin $B_{12}$ and red cell folate levels. Due to lack of reliable clinical cutoffs for these micronutrients during pregnancy for our population, we prefer to use the distribution within population to classify participants. Such an approach has been used earlier to report metabolic outcomes in the offspring in this cohort.[24] Distribution of outcome variables such as neurocognitive performance scores will be visualised by using different plotting techniques such as box plot and scatter plot. Univariate analysis (Pearson' correlation and linear regression) will be performed to examine the relationships between the primary exposures, life course variables and neurocognitive outcomes (table 1). We expect to construct multivariate linear (if outcome is continuous) or logistic regression (categorical

outcome) models to examine associations of our primary exposures with neurocognitive outcomes and extend the models to examine potential modifying and mediating effects of life course variables on the associations. Finally, we will adjust for the vitamin $B_{12}$ intervention status (vitamin $B_{12}$ intervention received yes or no) in the PRIYA trial. We will follow this broad scheme of analysis. Depending on observed univariate associations we may construct additional multivariate models.

Given the multitude of factors and possible collinearity, we will use dimensionality reduction techniques (such as principal component analysis or Lasso). The components might be used to construct path analytical models for exploring associations between outcomes.

### Objective 2

This examines the association between maternal nutritional exposures and brain structural functional connectivity:

1. Structural brain MRI analysis
   a. Voxel-based morphometry on T1 MPRAGE anatomical images will be performed using statistical parametric mapping in Matlab. Preprocessing steps will involve coregistration, segmentation. Spatial normalisation will transform all subject data to the same stereotactic space. Grey matter images will be smoothed by convolving with an isotropic Gaussian kernel. Statistical analysis using general linear model will be used to identify regions of grey matter concentration that are significantly associated with our exposures of interest.
   b. Cortical volume, thickness and hippocampal segmentation: A surface-based morphometric technique will be employed to measure cortical volume and thickness. We will use FreeSurfer software for image analysis. Standard steps include intensity normalisation, skull stripping and removal of non-brain tissues followed by segmentation, topology correction, registration and automated parcellation. Cortical volume and thickness will be computed for 68 regions (34 in each hemisphere) using the Desikan-Kelliany Atlas. The hippocampal subfield segmentation tool in FreeSurfer will be employed to compute volumes of the various hippocampal structures as per the inbuilt atlas which defines 11 hippocampal subfield structures.
2. Structural connectivity—diffusion tractography analysis: Diffusion-weighted images will be analysed using the 'The Diffusion Toolbox' software tool in the FMRIB Software Library (FSL; http://www.fmrib.ox.ac.uk/fsl/fdt/index.html). After preprocessing including motion correction and eddy current correction, the DTI indices fractional anisotropy, mean diffusivity (MD) and radial diffusivity will be computed. Structural connectome generation will be performed on the MRItrix3 software. Preprocessing steps will involve denoising and correction for eddy currents. The b0 volume of preprocessed images will be extracted and

registered to the T1 image using FSL's BET tool. The fibre orientation dispersion functions (FODs) will be generated for white matter and normalised. Probabilistic streamlines tractography will be performed by dynamically seeding the normalised white matter FOD's. The whole brain tractogram thus generated will be used for structural connectome generation.

3. Functional connectivity: CONN software (https://web.conn-toolbox.org/) will be used for the analysis of rsfMRI data. This will involve standard preprocessing steps such as realignment, normalisation to MNI space, elimination of temporal linear trends and regressing out nuisance covariates such as residual motion and physiological artefacts. aCompCor will be used for minimising physiological artefacts. Motion censoring and interpolation will also be used to remove the effects of large frame wise displacements due to head motion. Following preprocessing, mean time series will be extracted from 91 cortical and 15 subcortical regions in the FSL Harvard-Oxford atlas, as well as from 26 cerebellar regions in the AAL atlas. These time series will be used in a whole brain functional connectivity analysis employing Pearson's correlation coefficient between the mean time series.

For each modality, the following statistical comparisons will be performed. We will first perform correlations between maternal vitamin $B_{12}$ and folate exposures with brain MRI outcomes to examine for any linear relationships. We will use an analysis of covariance model to examine differences in MRI outcomes between low and high maternal vitamin $B_{12}$ groups. Similarly, we will examine for differences between low and high maternal red cell folate groups (selected based on median value of maternal red cell folate) and for any interaction between maternal vitamin $B_{12}$ and folate status. Variance due to subjects' age, gender, total intracranial volume, ACEs will be regressed out of the analyses. Effects of additional factors (such as current vitamin $B_{12}$ status of offspring, education status, socioeconomic status) will be examined.

For making inferences from multimodal imaging data, we will adopt a hierarchical approach. Accordingly, we will perform voxel-wise tests for difference in grey matter volume (or cortical thickness). For brain regions which are significant, DTI-based structural connectivity and fMRI-based functional connectivity between those regions will be investigated. This strategy assures that the effects of interest have both structural and functional aetiologies, and hence are likely to be robust and reproducible.

## Objective 3

This examines causality of association between maternal nutritional factors and neurocognitive outcomes. The genetic determinants of vitamin $B_{12}$ have been described in our population using GWAS,[36] which will be used to calculate a gene risk score of vitamin $B_{12}$ deficiency. Mendelian randomisation analysis (triangulation and instrumental variable analysis) using the genetic risk score for vitamin $B_{12}$ deficiency and homocysteine (MTHFRrs1801133) will be performed.

## Patient and public involvement

The present study is a follow-up of participants of the PMNS birth cohort. This cohort was set up in 1993 from six villages within Pune district in India. The cohort has been serially evaluated every 6 years (at 6, 12, 18 years) and is currently in the 24th year of its follow-up. The cohort participants were also part of a randomised controlled trial from 2012 to 2019. We are also following up the children born in the next generation of the cohort. The cohort has a high follow-up rate of 90% over the last 18 years. This was possible due to active community engagement and involvement of social workers who are from the same community and interact with the cohort participants regularly. Information on results of our studies is shared with cohort participants through public engagement initiatives.

## Ethics and dissemination

The study will be carried out as per the ethical guidelines provided by the Indian Council of Medical Research. The study has been approved by the Institutional Ethics Committee of the KEM Hospital Research Center, Pune (KEMHRC/RVM/EC/1249). The cohort participants will be approached by the field workers who have been following them in the community for last many years. Participants will be provided the study information sheet and an opportunity to discuss their doubts and concerns regarding the study with the investigator and then recruited after they sign a written informed consent.

The data generated from this study will be shared with the scientific community at local, national and international conferences, and published in open access peer-reviewed journals. The major findings will be shared with the general public, study participants and their families in lay terms through public engagement initiatives.

The study will help build a database on neurocognitive and mental health outcomes and brain imaging in young adulthood in a preconceptional birth cohort. The results of the association of maternal vitamin $B_{12}$ during the pregnancy and long-term neurodevelopmental outcomes will inform public health actions for nutritional health of adolescent girls and women of reproductive age.

**Author affiliations**
[1]Diabetes Unit, KEM Hospital Research Centre, Pune, Maharashtra, India
[2]AU MRI Research Center, Department of Electrical and Computer Engineering, Auburn University, Auburn, Alabama, USA
[3]Department of Psychological Sciences, Auburn University, Auburn, Alabama, USA
[4]Center for Neuroscience, Auburn University, Auburn, Alabama, USA
[5]Strategic Consulting, Cytel Inc, Cambridge, Massachusetts, USA

**Contributors** RVB and CY conceptualised and designed the study and wrote the manuscript. SKB designed the statistical analysis plan and power estimation. GD contributed to the brain imaging methods and analysis and framing the study objectives. All authors read and approved the final manuscript.

**Funding** This work is supported by the DBT/Wellcome Trust India Alliance Fellowship (IA/CPHI/161502665) awarded to RVB.

**Competing interests** None declared.

**Patient consent for publication** Not required.

**Provenance and peer review** Not commissioned; externally peer reviewed.

**ORCID iDs**
Rishikesh V Behere http://orcid.org/0000-0002-2395-0792
Chittaranjan Yajnik http://orcid.org/0000-0002-2911-2378

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
