## [Reviewer comments · BMJ Open]

ARTICLE DETAILS

TITLE (PROVISIONAL)	Maternal vitamin B12, folate during pregnancy and neurocognitive outcomes in young adults of the Pune Maternal Nutrition Study (PMNS) prospective birth cohort: Study protocol
AUTHORS	Behere, Rishikesh; Deshpande, Gopikrishna; Bandyopadhyay, Souvik; Yajnik, Chittaranjan

VERSION 1 – REVIEW

REVIEWER	Monk, Catherine Columbia University Medical Center
REVIEW RETURNED	24-Jan-2021

GENERAL COMMENTS	This manuscript describes a study protocol aiming to follow up adults' neurocognitive functioning in relation to their mothers intake of B12 and folate during pregnancy. Overall, this is a well-designed and straight forward study protocol. Some aspects of background could be strengthened to improve study justification and some small changes in methods (I think doable ones) are suggested, all below. 1. B12 and folate are distinct micronutrients yet in background they are quickly linked together, separated by a dash; prior to this pls include more info on properties of each or justify lumping together; describe how typically adequate levels are obtained in diets. Are they always correlated? After description of animal studies unique findings for each micro nutrient is given. Suggest edit so first distinct associations, then justify look at both together2. Is it really causality in Avon study?3. Oh, so focus of study is the imbalance of B12 and folate; background needs to better justify this approach, and set up that focus as an aim of the study. As it reads now, again, two micronutrients are lumped together, only a little info is given on significance of imbalance4. Page 8, line 5 on; summary of rationale for study is too broad as project will not look at nutrition overall but at two micronutrients (at least at this point that is what the reader believes)5. Page 9 controlling for does not list maternal prenatal stress though in the background this was listed as a weakness in prior studies6. Aim 3 causality of association is not consistent with methods of study which is observational/associations
--

	7. Page 10 row 25, the info on F1 participants randomized to nutrition intervention is given w/o explanation for how this may impact this study's findings; this is finally described in statistics section, would be helpful to have sooner 8. It is not clear how B12 and Folate are measured during pregnancy? 9. Would be important to estimate % in sample who will be high folate/low B12 and possible the other combinations; also, aren't there clinical cut points for these micronutrients in pregnancy that could be used to define exposure? 10. Need more specifics on statistical modeling of how exposure variable will be defined and constructed; overall more clarity needed on how exposure variable is being defined and statistically constructed 11. Imaging aims are quite broad; pls target specific brain regions relevant to hypotheses esp for functional data
--	--

REVIEWER	Hoffman, Daniel Rutgers university, Nutritional Sciences
REVIEW RETURNED	28-Jan-2021

GENERAL COMMENTS	This paper describes a very well conducted study on early life experiences and cognition based on data collected in a large cohort in India. The authors present their research well, but there are a number of major issues that must be revised before a final recommendation can be made. 1. The Introduction suffers from cursory writing and does not flow well. The authors are requested to revise the Introduction to make each paragraph more clear and connect to the prior and paragraph in a logical manner rather than simply writing summaries of studies. Line 40: Change "Avon's" to "Avon" Line 50: Change "In the Indian setting..." to "In India, ..." Line 54: Please be specific as to what "It.." is as it not clear if the policy is the problem or supplementation. Line 58: Please be specific as to what "This..." is referring to. From the Introduction to the end of the paper is where the most serious problems are located. In short, the text appears to be copied directly from a grant application as the format and wording is more consistent with planned work rather than completed research. The outline clearly names objectives and specific aims, items most commonly found in a grant application and not in research papers. Moreover, the use of the future tense and the high level of detail in the methods is not consistent with journal articles. These sections must be revised before a final recommendation can be made. While these issues are important and must be revised, there is still great enthusiasm for this research and the authors should revise the paper accordingly.
---

VERSION 1 – AUTHOR RESPONSE

Reviewer: 1

Dr. Catherine Monk, Columbia University Medical Center

1. B12 and folate are distinct micronutrients yet in background they are quickly linked together, separated by a dash; prior to this pls include more info on properties of each or justify lumping together; describe how typically adequate levels are obtained in diets. Are they always correlated? After description of animal studies unique findings for each micro nutrient is given. Suggest edit so first distinct associations, then justify look at both together

We thank the reviewer for this suggestion. We have restructured the introduction section to focus on our main research question i.e. the effect of maternal micronutrients vitamin B12 and folate on long term cognitive outcomes in the offspring. As suggested we first describe the role of B12 and folate independently and then potential adverse effects of a B12 folate imbalance.

2. Is it really causality in Avon study?

Reference to causality has been removed in the introduction section (page 5 para 1).

“The Avon’s Longitudinal Study on Parents And Children (ALSPAC) reported association between low maternal dietary intake of vitamin B12 and lower IQ scores at 8 year and genetic determinants of B12 (FUT2gene)”

3. Oh, so focus of study is the imbalance of B12 and folate; background needs to better justify this approach, and set up that focus as an aim of the study. As it reads now, again, two micronutrients are lumped together, only a little info is given on significance of imbalance

Thank you for the suggestion. We have restructured the introduction to highlight the significance of the imbalance and justify the approach.

4. Page 8, line 5 on; summary of rationale for study is too broad as project will not look at nutrition overall but at two micronutrients (at least at this point that is what the reader believes)

The rationale has been modified to focus on the two micronutrients B12 and folate.

5. Page 9 controlling for does not list maternal prenatal stress though in the background this was listed as a weakness in prior studies

Information on maternal prenatal stress is not available on our cohort.

6. Aim 3 causality of association is not consistent with methods of study which is observational/associations

The Aim 1 & 2 are to examine associations. In Aim 3 we plan to test the causality of these associations using information on genetic determinants of Vitamin B12 (FUT2, FUT6, TCN 2 in mother and child) by a mendelian randomization approach. This has been elaborated under objective 3 under Aims section and Plan of data analysis section.

7. Page 10 row 25, the info on F1 participants randomized to nutrition intervention is given w/o explanation for how this may impact this study’s findings; this is finally described in statistics section, would be helpful to have sooner

“The effects of later life intervention will be adjusted during statistical analysis.” This statement has been added along with description of nutrition intervention.

8. It is not clear how B12 and Folate are measured during pregnancy?

The relevant information has been added under methods -page 8 para 1

“Maternal micronutrients vitamin B12, red cell folate and ferritin have been measured from blood samples collected at 18 and 28 weeks of gestation. Vitamin B12 levels were measured using the microbiological assay technique and red cell folate and ferritin by radioimmunoassay”

9. Would be important to estimate % in sample who will be high folate/low B12 and possible the other combinations; also, aren't there clinical cut points for these micronutrients in pregnancy that could be used to define exposure?

Clinic cut points are developed for western populations who have very different nutritional exposures compared to Indians further complicated by the fact that many western populations have been exposed to folic acid fortification in last two decades. The PMNS (1993-96) is a 'pristine' community based cohort representing natural nutritional status. The only intervention was the IFA tablets (100tablets given to pregnant mothers at ~ 18 weeks gestation by the research team). Therefore we have preferred to use distribution within population to classify participants and also continuous analysis. Such an analysis has given rewarding results for metabolic outcomes (Yajnik et al 2008).

10. Need more specifics on statistical modeling of how exposure variable will be defined and constructed; overall more clarity needed on how exposure variable is being defined and statistically constructed

Our primary exposure of interest is maternal vitamin B12 and red cell folate levels. Due to lack of reliable clinical cutoffs for these micronutrients during pregnancy for our population, we prefer to use the distribution within population to classify participants. Such an approach has been used earlier to report metabolic outcomes in the offspring in this cohort (Yajnik et al 2008). We have updated the plan of analysis section with these details (page 13).

11. Imaging aims are quite broad; pls target specific brain regions relevant to hypotheses esp for functional data

We propose to examine hippocampal and frontal-subcortical connectivity in relation to nutritional exposures. This has been stated in the hypothesis section. In the introduction, we provide background information that would motivate such a hypothesis:

“Early intrauterine insults can impact development of brain areas that show high neuronal growth such as hippocampus and cortical areas (4). Later life factors influence synaptic pruning and myelination (5). Deranged Brain structural and functional connectivity within and between hippocampal, frontal and subcortical networks is known to be associated with impaired neurocognitive functioning and neurodevelopmental disorders (6,7).” – Introduction page 4

Reviewer: 2

Dr. Daniel Hoffman, Rutgers university

Comments to the Author:

This paper describes a very well conducted study on early life experiences and cognition based on data collected in a large cohort in India. The authors present their research well, but there are a number of major issues that must be revised before a final recommendation can be made.

1. The Introduction suffers from cursory writing and does not flow well. The authors are requested to revise the Introduction to make each paragraph more clear and connect to the prior and paragraph in a logical manner rather than simply writing summaries of studies.

Line 40: Change "Avon's" to "Avon"

Line 50: Change "In the Indian setting..." to "In India, ..."

Line 54: Please be specific as to what "It.." is as it not clear if the policy is the problem or supplementation.

Line 58: Please be specific as to what "This..." is referring to.

The Introduction section has been rewritten to improve the flow with a focus on the proposed research question. The suggested line changes have been incorporated.

From the Introduction to the end of the paper is where the most serious problems are located. In short, the text appears to be copied directly from a grant application as the format and wording is more consistent with planned work rather than completed research. The outline clearly names objectives and specific aims, items most commonly found in a grant application and not in research papers. Moreover, the use of the future tense and the high level of detail in the methods is not consistent with journal articles. These sections must be revised before a final recommendation can be made.

This paper describes protocol of methods of proposed work and is not completed research. In keeping with the journal guidelines for protocols the manuscript has been written in the future tense(<https://bmjopen.bmj.com/pages/authors/>). We have verified with the editor that this format indeed conforms with what is expected from the journal.

We followed the format as in previously published protocol papers in this journal which describes proposed methods and has used future tense in the manuscript.

(eg. Anand S, Thomas S, Jayachandra M, et al. Effects of maternal B12 supplementation on neurophysiological outcomes in children: a study protocol for an extended follow-up from a placebo randomised control trial in Bangalore, India. *BMJ Open* 2019;9:e024426. doi:10.1136/ bmjopen-2018-024426).

We have edited out the specific details of methods (behavioural assessments) in the main text to make it concise. Specific details are now provided as supplementary material.

While these issues are important and must be revised, there is still great enthusiasm for this research and the authors should revise the paper accordingly.

We thank the reviewer for this comment and we are submitting the revised manuscript.

VERSION 2 – REVIEW

REVIEWER	Monk, Catherine Columbia University Medical Center
REVIEW RETURNED	29-May-2021
GENERAL COMMENTS	This study protocol report proposes to investigate associations between maternal micronutrient intake during pregnancy

	(specifically B12 and folate) in relation to offspring neurocognitive development at age 12. There are many strengths to this protocol including the large sample size, prospective design, stored biological samples and outcomes that include behavior (some objectively observed, e.g., IQ) and behavior indices. Overall, the protocol is very strong with only several minor weaknesses that are readily addressable.  1. Abstract, first comma in first sentence not needed 2. Page 4, line 22 deranged is an unusual word/not typical for medical/science writing 3. Page 6, line 32, NTD not yet defined 4. Sometimes background moves between studies on low B12/high folate, or low on both, yet rationale for study seems to be focused on imbalance between the two. Greater consistency in reporting these supportive studies and/or acknowledging the differences in background studies to one proposed would enhance study's rationale. 5. Aims and objective broadly stated but again, background seems to focus on imbalance in two micronutrients. 6. Primary objective #2 should provide more specific outcomes; less or more gray matter volume, etc? Yes, some of this is given under hypotheses, but more specificity is needed; operationalize the result of 'adversely impacting development', greater or less connectivity? 7. Are the estimates for recruitment still valid in the context of the 2019 COVID pandemic? 8. Resting state functional connectivity with participants 'not thinking about anything particular' should be justified versus more controlled approach with stimuli presented to ensure more common mental/psychological state across participants. 9. Objective 3 (throughout) needs more background for its justification and better explanation. 10. Consider dichotomizing outcomes by sex.
--	--

VERSION 2 – AUTHOR RESPONSE

Reviewer: 1

Dr. Catherine Monk, Columbia University Medical Center, Columbia University

1. Abstract, first comma in first sentence not needed

Change incorporated

2. Page 4, line 22 deranged is an unusual word/not typical for medical/science writing

Deranged replaced with 'altered'

3. Page 6, line 32, NTD not yet defined

NTD has been expanded

4. Sometimes background moves between studies on low B12/high folate, or low on both, yet rationale for study seems to be focused on imbalance between the two. Greater consistency in reporting these supportive studies and/or acknowledging the differences in background studies to one proposed would enhance study's rationale.
5. Aims and objective broadly stated but again, background seems to focus on imbalance in two micronutrients.

We have reorganized the introduction as per the reviewers suggestions. We first highlight the important role of maternal B12 and folate in offspring neurodevelopment through its effect on the one carbon metabolism pathway. We review evidence from observational studies on associations of maternal B12 or folate on cognitive outcomes. Next we describe the Indian context where a folic acid supplementation in a population with widely prevalent B12 deficiency leads to a B12/folate imbalance. We provide evidence based on animal studies and observations in our cohort for metabolic outcomes. We then propose our study question and hypothesis. The hypothesis is elaborated in response to reviewer comment no. 6 below

6. Primary objective #2 should provide more specific outcomes; less or more gray matter volume, etc? Yes, some of this is given under hypotheses, but more specificity is needed; operationalize the result of 'adversely impacting development', greater or less connectivity?

Response: The specification of hypotheses have been expanded to add more specificity as follows: "We hypothesize that a lower maternal B12, higher folate, higher homocysteine nutrient pattern during pregnancy adversely impacts development of brain regions. This will be reflected in brain imaging as reduced brain volume as measured through anatomical images, weaker connections in structural/functional networks (hippocampal and frontal-subcortical connections) involved in cognitive processing (memory, executive functions) as measured through DTI and resting state fMRI, respectively. These structural and functional changes will in turn result in poorer neurodevelopmental outcomes (lower neuropsychological test performance, adverse temperamental traits and higher risk for common mental disorders) in the young adult offspring."

7. Are the estimates for recruitment still valid in the context of the 2019 COVID pandemic?

The study protocol was approved for funding in 2017. Subject recruitment was initiated in May 2018 and had to be halted in March 2020 due to the COVID-19 pandemic. As of March 2020 neurocognitive assessments have been performed on 386 participants of the cohort out of the intended 550 participants and Brain MRI scans obtained on 195 participants out of the intended 200. We will try to recruit more participants once the situation is conducive. We acknowledge that due to the COVID-19 pandemic there will be some reduction in the power of the study. The results will be reported as per emerging guidelines on reporting clinical studies affected by the COVID-19 pandemic (Perlis et al 2021).

We have added the above section on 'current status and impact of COVID-19 pandemic' in the revised manuscript.

We provide estimated power calculations at different recruitment rates. Assuming an association of 0.11 standard deviation of continuous outcome at 5% significance level.

Sample size	Estimated power
386	70%
450	75%
500	79%
550	83%

8. Resting state functional connectivity with participants 'not thinking about anything particular' should be justified versus more controlled approach with stimuli presented to ensure more common mental/psychological state across participants.

Response: We believe this question has been answered by more than a decade of research in fMRI and the utility of resting state fMRI for assessing brain function in health and disease is quite undisputed in the field today.

When human subjects get scanned while letting their mind wander, as in resting state fMRI, it characterizes the "default mode of brain function" (Buckner 2008, Raichle 2015). The state of the brain in the absence of focused attention and task is captured by the correlational structure in spontaneous activity, i.e. resting state brain networks. As opposed to task-based fMRI which characterizes evoked responses in specific brain systems, these resting state brain networks encompass the entire brain. When comparing brains of two populations where in one expects behavioural differences in tasks between the groups, it is difficult to attribute task fMRI activation differences to the differences between groups alone because it could also be driven by differences in task performance. This problem is inherent when one is looking at evoked responses. This can be overcome by studying spontaneous responses as in resting state fMRI.

The interesting fact is that resting state brain networks seem to correspond rather well with brain systems evoked by tasks (Stephen et al 2009). Therefore, it seems convenient to use one resting fMRI scan to characterize brain-wide networks rather than a battery of tasks to engage different brain systems. Further, a large body of literature suggests that resting fMRI is far better than task fMRI for predicting disease state or behaviour in human subjects (Fleisher et al 2009, Deshpande et al 2010). We certainly do not argue that

task fMRI is without its merits. Task fMRI provides evoked response whereas resting fMRI characterizes spontaneous response. They are materially different types of information. However, when time available for fMRI within a scan is limited (as it is in our case, as also in many common situations), the current consensus in the field suggests that it is better to acquire resting state fMRI data rather than task fMRI data.

A justification for resting state fMRI has been added under the image acquisition section in the revised manuscript.

9. Objective 3 (throughout) needs more background for its justification and better explanation.

The objective 1 & 2 assess associations between intrauterine B12 and folate exposures with neurocognitive outcomes. However associations do not support causality. Causality can be answered by performing randomized controlled trials but these require considerable time and effort and funding and there are ethical considerations. The technique of Mendelian randomization allows us to test causality of observed associations using genetic determinants of B12 and folate status. Earlier studies have used this technique with genetic determinants of B12 (FUT2 and TCN2) to examine associations with IQ in offspring at 8 years in the ALSPAC cohort) (Bonilla et al 2012). We have earlier reported a mendelian randomization study using genetic determinants of maternal homocysteine (MTHFR) to test the causality of association between maternal homocysteine and low birth weight in the offspring (Yajnik et al 2014). The genetic determinants of B12 have been described in our population using GWAS (Nongmaithem et al 2017). Therefore we are able to calculate a gene risk score of B12 deficiency which will be a powerful genetic tool to study causality. A background to objective 3 has been added in the revised manuscript.

10. Consider dichotomizing outcomes by sex.

Yes we will also examine outcomes by gender